# Micronized Shell-Bioaggregates as Mechanical Reinforcement in Organic Coatings

**DOI:** 10.3390/ma17164134

**Published:** 2024-08-21

**Authors:** Francisco Javier Rodríguez-Gómez, Massimo Calovi, Stefano Rossi

**Affiliations:** 1Departamento de Ingeniería Metalúrgica, Facultad de Química, Universidad Nacional Autónoma de México, Del. Coyoacán, Mexico City 04510, Mexico; fxavier@unam.mx; 2Department of Industrial Engineering, University of Trento, Via Sommarive 9, 38123 Trento, Italy; massimo.calovi@unitn.it

**Keywords:** bio-aggregates, shells, mechanical properties, abrasion, organic coatings

## Abstract

Shells are primarily composed of calcite and aragonite, making the inclusion of micronized shells as bio-based fillers in organic coatings a potential means to enhance the mechanical properties of the layers. A water-based coating was reinforced with 5 wt.% *Acanthocardia tuberculata* powder, 5 wt.% *Mytilus galloprovincialis* powder, and 5 wt.% of an LDPE/ceramic/nanoceramic composite. An improvement in abrasion resistance was achieved using micronized seashells, as demonstrated by the Taber test (evaluating both weight loss and thickness reduction). Additionally, Buchholz hardness improved with powders derived from *Mytilus galloprovincialis*. No significant differences were observed among the samples in terms of color and gloss after 200 h of UV-B exposure. However, the delamination length from the scratch after 168 h of exposure in a salt spray chamber indicated that the addition of particles to the polymeric matrix resulted in premature degradation, likely due to the formation of preferential paths for water penetration from the scratch. This hypothesis was supported by electrochemical impedance spectroscopy measurements, which revealed a decrease in total impedance at 0.01 Hz shortly after immersion in a 3.5% NaCl solution. In conclusion, the particle size and shape of the micronized shells improved abrasion resistance without altering color and gloss but led to a decrease in the coating’s isolation properties.

## 1. Introduction

Wear resistance of metallic surfaces can be achieved through the use of resistive coatings and lubricants. However, in scenarios where lubricants are not allowed, the protective coating must include some form of lubricant to maintain their performance. Industrial applications require abrasion-resistant coatings with high mechanical properties, such as hardness and toughness [1]. Reinforcing an organic matrix to improve abrasion protection has been accomplished by adding ceramic particles (e.g., Al_2_O_3_, SiO_2_) [2,3,4] or polymeric aggregates, which act as lubricants [5]. Even in small quantities, ceramic particles can significantly enhance the wear resistance of a coating. Micronized particles of ceria or zirconia [6,7] have been used to boost the mechanical and protective properties of coatings, enhancing substrate protection (anticorrosive properties) without negatively affecting aesthetic properties like color or gloss. This objective must be met using environmentally friendly materials. Nevertheless, the use of bio-based materials often poses significant challenges related to the protective performance, durability, and color consistency of additives in the paint [8].

The sea represents a significant resource of bio-based fillers that can be effectively utilized. Molluscs’ seashell waste, abundant in calcium carbonate (CaCO_3_) as its major component, offers a renewable alternative to conventional sources like calcium carbonate. Specifically, marine shells such as *Acanthocardia tuberculata* and *Mytilus galloprovincialis* are predominantly composed of calcite and aragonite [9]. Calcite derived from these shells can serve as a filler in organic coatings and has been studied for its potential to reduce the need for TiO_2_ [10]. Utilizing calcite from marine shells not only reduces coating costs but also enhances mechanical properties. Research has explored using marble residues [11] to promote circular economy practices and employ environmentally sustainable materials.

Recent studies have shown that calcium carbonate obtained from seashells can enhance the mechanical [12,13] and thermal [14] properties of bio-composites. Additionally, it improves the durability and enhances flame retardancy and smoke suppression in coatings [15,16]. These findings underscore the multifaceted benefits of utilizing seashell-derived calcium carbonate in various applications, highlighting its potential as a valuable bio-based material. Similarly, bio-aggregates reinforcing cementitious matrices have been enhanced with the addition of marine shells [17,18]. The literature indicates that the lamellar shape of micronized seashell powder and its chloride content influence adhesion when included in a cementitious matrix [19]. K. Hasan et al. [20] reported a significant decrease in water permeability with the addition of seashell ashes up to 25% of the cement paste, likely due to pore sealing, which impedes water and contaminant ingress; up to 15% replacement was deemed optimal for porosity and absorption. C. Martínez-García et al. [21] found that incorporating mussel sand (1 mm average particles) in concrete up to 25 wt.% resulted in good performance by modifying pore size and shape.

Given the promising potential of seashell powder for enhancing the mechanical properties of various materials without adversely affecting porosity, water uptake, or aesthetics, this research explores the addition of micronized marine shells in an acrylic-polyurethane matrix. The study focuses on evaluating the reinforcement effect of seashells- and mussels-derived powders applied as bio-based fillers into a water-based organic coating to improve its mechanical properties, specifically, abrasion resistance and hardness. The protective features of the coatings were assessed through UV exposure, salt spray chamber tests, and electrochemical impedance spectroscopy, with measurements of color and gloss retention and variations in electrochemical parameters over time. To demonstrate the applicability of these bio-based fillers, the performance of the two powders derived from marine waste was compared with another commercially available *green* additive specifically designed as a reinforcing filler for paints.

## 2. Materials and Methods

### 2.1. Materials

The composite filler *PolyTuf^®^ 1229*, provided by Micro Powders (Tarrytown, NY, USA), was utilized in its original form. This composite material consists of low-density polyethylene (LDPE) and alkali aluminosilicate ceramic microspheres. It appears white in color, has a melting point between 110 and 113 °C, a density of 0.97 g/cc, and an average particle size ranging from 9 to 12 µm. Otherwise, the two bio-based powders were produced by grinding seashells (*Acanthocardia tuberculata*) and mussels (*Mytilus galloprovincialis*) collected from the beaches of the Adriatic Sea. Specifically, an MGS ball mill (MGS, Fiorano Modenese, Italy) was used to grind the powders. The process utilized aluminous porcelain spheres with a diameter of 1 cm, which were stirred inside the jar containing the powder for a duration of 10 min. Each material was ground separately and then sieved to achieve a micrometric powder. Acetone and NaCl were purchased from Sigma-Aldrich (St. Louis, MO, USA) and used as received. The carbon steel substrate, specifically a Q-panel type R (0.15 wt.% C-Fe bal.) measuring 152 mm × 76 mm × 0.8 mm, was sourced from Q-lab (Westlake, OH, USA). Additionally, the waterborne 2K polyurethane-acrylate paint, *IDROPUR ZW 01*, was provided by EP Vernici (Solarolo, Ra, Italy). It has a dry residue of 35–37 wt.%, specific weight of 1000–1050 g/L and transparent appearance. This specific polyurethane-acrylate paint was used for several reasons. First, as a clear coat, it is free of additives and pigments that could reduce or influence the effect of the bio-based fillers characterized in the study. Additionally, the transparency of the paint facilitates the observation of the fillers introduced into it. Finally, its good outdoor durability allows for the evaluation of the fillers’ behavior in accelerated degradation tests.

### 2.2. Samples Production

Prior to applying the paint, the metal substrates were cleaned with acetone to eliminate any surface grease, ensuring optimal adhesion to the polymer matrix. Four sets of samples were produced and characterized in this study, as outlined in Table 1. Sample R, which served as a reference in all analyses, was created by spraying the industrial paint. In contrast, the other three sets of samples involved modifying the polyurethane-acrylate paint by adding 5 wt.% of a reinforcing filler. Specifically, *PolyTuf^®^ 1229* powder was added to create the coating for sample P, while powders derived from seashells and mussels were used in the formulations for samples S and M, respectively. To ensure homogeneous mixtures and even distribution of the powders within the polymer matrix, the paint mixtures were mechanically stirred for 30 min prior to application. Thus, the various paint formulations were applied using a spray method with pressure of 3 bar, achieving a coverage rate of approximately 8 m^2^/L. After the application of the paint, the coatings were cured by heat treatment in an oven at 60 °C for 40 min, following the recommendations of the paint manufacturer.

### 2.3. Characterization

Initially, the appearance of the three types of bio-based additives was examined using an optical stereomicroscope (Nikon SMZ25, Nikon Instruments Europe, Amstelveen, The Netherlands). Their morphology was further investigated in detail using a low vacuum scanning electron microscope (SEM JEOL IT 300, JEOL, Akishima, Tokyo, Japan) with an acceleration voltage of 15.0 kV and a working distance (WD) of approximately 10.0 mm. Additionally, Energy-dispersive X-ray spectroscopy (EDXS, Bruker, Billerica, MA, USA) was employed to analyze their chemical composition.

Furthermore, the aesthetics of the coatings were examined through colorimetric and gloss analysis to assess the impact of the various types of bio-based fillers. Colorimetric analyses were performed using a Konica Minolta CM-2600d spectrophotometer (Konica Minolta, Tokyo, Japan) with a D65/10° illuminant/observer configuration in SCI mode. Gloss measurements were conducted according to the ASTM D523/14 standard [22] using an Erichsen 503 instrument from Erichsen Co.Fo.Me.Gra Instruments (Milan, Italy). Then, color and gloss assessments were carried out on 5 samples per series, with 3 measurements per sample. Additionally, surface roughness analyses were performed in conjunction with the gloss measurements to evaluate how surface texture influences the coatings’ aesthetics. These measurements were carried out using the MarSurf PS1 mobile surface roughness measurement instrument (Carl Mahr Holding, Göttingen, Germany). Moreover, the appearance of the coatings was evaluated using both SEM and optical stereomicroscope to examine how the different additives impact the structure of the composite layers.

The impact of the bio-based additives on the mechanical properties of the polymer matrix was assessed through hardness and abrasion resistance tests. The Buchholz hardness indentation test was conducted according to the ISO 2815 standard [23], using an Elcometer 3095 Buchholz Hardness Tester (Elcometer, Manchester, UK). A consistent testing force of 0.5 kg was applied for 30 s on the coating surface, and the depth of the impression made by the standardized instrument indicated the coatings’ hardness was measured. Five measurements were taken for each sample. For abrasion resistance, the Taber abrasion test was performed using a TABER 5135 Rotary Platform Abrasion Tester (Taber Industries, North Tonawanda, NY, USA), following ASTM D4060-10 standard guidelines [24]. This test involved two CS17 abrasion wheels with a combined weight of 0.5 kg, subjecting the samples to 1000 Taber cycles while measuring mass loss. After the test, the surfaces of the samples were examined using SEM to investigate the role of bio-based fillers in mitigating abrasion phenomena.

The impact of the three types of fillers on paint durability was evaluated through different accelerated degradation tests. The samples were exposed to a salt spray chamber (Ascott Analytical Equipment Limited, Tamworth, UK) for 168 h, following ASTM B117-11 standards [25], using a 5 wt.% sodium chloride solution. This test aimed to assess how the bio-based fillers influence the corrosion protection performance of the coatings in an aggressive environment. Additionally, a mechanical scratch was applied to the surface of the painted samples to evaluate coating adhesion.

Furthermore, the samples were subjected to UV-B radiation for 200 h to simulate outdoor exposure. This was done using a UV173 Box Co.Fo.Me.Gra (Co.Fo.Me.Gra, Milan, Italy) in accordance with ASTM D4587-11 standards [26]. The samples were exposed to UV-B radiation (313 nm) at 60 °C. Throughout the exposure period, colorimetric and gloss analyses were conducted to assess the effect of the additives. Although this test uses strong UV-B radiation to mimic some of the solar radiation the samples might encounter, it was chosen to highlight potential physical and chemical degradation of the bio-based additives. The exposure was terminated after 200 h when no significant degradation was noted and the color change appeared to stabilize.

Lastly, the protective properties of the coatings were evaluated using electrochemical impedance spectroscopy (EIS) with a Parstat 2273 potentiostat (Princeton Applied Research by AMETEK, Oak Ridge, TN, USA) and PowerSuit ZSimpWin (version 2.40) software. A signal with a peak-to-peak amplitude of approximately 15 mV was applied across a frequency range from 10^5^ to 10^−2^ Hz. The setup included an Ag/AgCl reference electrode (+207 mV SHE) and a platinum counter electrode, with samples immersed in a 3.5 wt.% sodium chloride aqueous solution for 168 h. The testing area analyzed was 6.5 cm^2^.

## 3. Results and Discussion

### 3.1. Fillers and Coatings Appearance

Figure 1 displays the visual characteristics of the three fillers, as seen through the stereomicroscope on the left and the SEM on the right. The stereomicroscope images highlight the aesthetic characteristics of the powders, whereas the SEM analyses provide valuable insights into their morphology and chemical composition. *PolyTuf^®^ 1229* (Figure 1a) is observed as a white powder. Detailed SEM analysis reveals that this powder comprises two components: organic matrix granules and ceramic spheres. The organic matrix granules exhibit an irregular structure with an average size of less than 10 µm, as indicated by the manufacturer. In contrast, the ceramic spheres are smaller, consistently measuring less than 5 µm in diameter. EDXS analyses, presented in Appendix A, suggest that the ceramic spheres are composed of alkali aluminosilicate. The seashell processing product (Figure 1b) also appears as a white powder, but with a distinctly different structure. The granules vary greatly in size, ranging from a few µm to over 60 µm. Although a sieve with a 32 µm mesh size was used, some elongated granules larger than 50 µm were still able to pass through. These granules appear as compact yet irregular materials, characteristic of shell by-products [27,28]. EDXS analyses (Appendix A) confirm that they are primarily composed of calcium carbonate, which constitutes 95% of the content of marine shells [29]. Mussel by-products, on the other hand, appear as a white-purple powder (Figure 1c). This filler also exhibits a highly irregular size distribution, with larger granules consisting of smaller powder particles compacted into a prismatic layer structure typical of mussel shells [30,31]. EDXS analyses (Appendix A) confirm the calcitic nature of the mussel by-product [32].

Therefore, the three types of fillers were incorporated into the polyurethane-acrylate paint to produce the four series of samples listed in Table 1. The coatings were initially evaluated for their aesthetic properties to determine how the powders influenced color and gloss. Figure 2 illustrates the results of the colorimetric measurements. Specifically, Figure 2a shows the values for the three color coordinates: L*, a*, and b*. These coordinates measure different color attributes:L* represents lightness, ranging from 0 (black) to 100 (white);a* denotes the red-green axis, with positive values indicating red and negative values showing green;b* reflects the yellow-blue axis, where positive values represent yellow and negative values denote blue.

Incorporating the three bio-based powders into the paint generally results in a decrease in the brightness of the coatings, with L* values falling from approximately 71 for the reference sample (R) to around 67 for the three composite coatings. However, it could be asserted that the fillers do not function as pigments, as the changes in the a* and b* coordinates are minimal. Thus, even though the mussel powders have a light purple hue, they do not seem to significantly affect the color of the transparent paint. Nevertheless, a small real chromatic effect is appreciable, as illustrated by the graph in Figure 2b, which depicts the color change (ΔE) of the three composite coatings relative to the reference sample R. This change was calculated using the equation provided in reference [33]:ΔE = [(ΔL*)^2^ + (Δa*)^2^ + (Δb*)^2^]^1/2^
(1)

Scientific research has shown that the human eye can detect color changes of 1 unit or more [34]. Therefore, the graph illustrates that the inclusion of bio-based fillers in the coatings leads to a significant color change, with ΔE values ranging from 4 to 5 units. This outcome is effectively attributed to a combined effect of reduced brightness (as the fillers obscure the surface) and slight increases in the a* and b* coordinates.

However, the most significant aesthetic impact of the fillers is evident in the reflectance properties of the coatings. Figure 3a presents the gloss measurement results, where the reference sample R displays very high gloss due to the paint’s highly reflective nature. Moreover, the elevated gloss levels are attributed to the paint’s transparency and lack of pigments, which enables the underlying highly reflective metal substrate to be prominently visible. In contrast, the addition of the three *green* fillers leads to a notable decrease in gloss, indicating that these additives function as mattifying agents. Since increased surface irregularity can reduce luminous reflectance, the samples were subjected to roughness measurements (Figure 3b) to ensure that the observed decrease in gloss is not due to changes in the surface morphology of the coatings caused by the presence of fillers. The graph addresses this concern: the surface roughness of all four series of samples is comparable. Consequently, the reduction in gloss is not linked to changes in surface morphology but rather to the mattifying effect of the bio-based fillers, which reduces the high reflectance of the polyurethane-acrylate paint.

The appearance of the samples is further detailed by the microscopic analyses shown in Figure 4. This figure presents the cross sections of the coatings as observed under the SEM (left) and top-view micrographs taken with the stereomicroscope (right). The reference sample R (Figure 4a) appears compact with a thickness of approximately 40 µm. The top-view image reveals a transparent layer that permits the underlying metal substrate to be visible. In contrast, the coating of sample P, as shown in Figure 4b, reveals a distinctly different structural morphology. Large voids are apparent within the coating, created by the LDPE powders from *PolyTuf^®^ 1229*, which were expelled due to brittle fracture in liquid nitrogen used to examine the sectioned cladding. Despite these voids, some ceramic microspheres remain visible, being smaller and firmly adhered to the polymer matrix. Another notable aspect is the increase in coating thickness, which reaches approximately 60 µm. This is attributed to the higher volume of material applied. Specifically, the addition of *PolyTuf^®^ 1229* powder at 5 wt.%—which has a lower density compared to the paint matrix—leads to a substantial increase in both the mass and volume of the sprayed material. Simultaneously, the top-view appearance of the coating also changes, showing the presence of the white powders incorporated within the paint polymeric matrix. Similar characteristics are observed in samples S and M (Figure 4c and Figure 4d, respectively). The translucent paint matrix reveals the presence of the fillers when viewed under an optical microscope, and these fillers are also clearly identifiable in the cross-sectional SEM analyses. Furthermore, in these cases, an increase in coating thickness is also observed, ranging from 50 to 55 µm. This increase is attributed to the addition of fillers in the paint formulation. SEM analyses reveal the presence of large granules of bio-based powders in both samples. These granules are not expelled during the brittle fracture process in nitrogen but are instead sectioned along with the coating. This observation indicates good adhesion and compatibility between the marine waste-derived powders and the paint matrix.

This aspect is particularly promising as it indicates that these powders have good potential as fillers in organic coatings. Rather than altering the color of the coating, these additives significantly affect its aesthetic reflectance properties by introducing opacifying effects. Ultimately, neither *PolyTuf^®^ 1229* nor the powders from seashells and mussels function as pigments. Instead, they serve as bio-based and eco-sustainable matting agents.

### 3.2. Impact of the Additive on the Coating’s Mechanical Characteristics

The samples were subsequently subjected to surface hardness and abrasion resistance tests, to evaluate the reinforcing effects of the bio-based additives. Specifically, the performance of powders derived from seashells and mussels was assessed to determine if they could serve as a natural alternative to the commercial *PolyTuf^®^ 1229*, which is designed explicitly as a reinforcing filler for paints.

The graph in Figure 5 reveals only minor variations in hardness among the four sample sets. It appears that adding fillers does not notably enhance the hardness of the paint’s polymeric matrix; in fact, the opposite may be true. For instance, the average indentation in sample P is greater than that of the reference R, indicating that the *PolyTuf^®^ 1229* additive is less hard compared to the standard reference matrix. Despite the inclusion of hard ceramic microspheres, the primary impact of *PolyTuf^®^ 1229* stems from the LDPE granules, which are softer than the polyurethane-acrylate resin in which they are embedded. A similar response is observed in sample S, where the large particles from seashells can be easily scratched and penetrated by the Buchholz indenter. Conversely, the performance of sample M is consistent with the reference sample R, indicating that the mussel powder has a negligible effect on the coating’s hardness. On the other hand, the literature demonstrates the difficulty of improving the hardness of paints using bio-based fillers [35,36], unless they are derived from very hard materials, such as macadamia nuts [37] or olive stones [38].

Despite the minimal impact on hardness, these fillers may exhibit different behaviors under abrasive conditions. Figure 6a illustrates the evolution of mass loss as a function of the number of Taber abrasion cycles. The results indicate a clear protective effect of the three types of bio-based additives. Sample R demonstrates a linear and consistent mass loss. In contrast, the three composite coatings show constant mass loss trends but with significantly reduced values. At the end of the test (1000 cycles), samples P, S, and M show reductions in mass loss compared to sample R by 75%, 67%, and 35%, respectively. Since the additives may have different densities compared to the paint matrix, these mass loss values are also associated with reductions in coating thickness, as depicted in Figure 6b. This comparison verifies that the reduced mass loss observed with the bio-based fillers corresponds to a similar trend in volume loss. Indeed, the trends are particularly similar, showing reduced thickness (and consequently volume) losses due to the presence of mussel powder, and more significantly, seashell and *PolyTuf^®^ 1229* powders. Therefore, the protective effect observed with the Taber measurements is not related to differences in system densities: the reduction in material loss, both in mass and volume, is effectively due to the protective properties of the bio-based additives.

To better emphasize the protective role of the fillers, the sample surfaces were examined under SEM after the Taber test. Figure 7 illustrates the morphological characteristics of the surfaces from the four series of samples subjected to abrasion. Sample R (Figure 7a) exhibits a typically irregular surface morphology caused by the shearing stresses imparted by the Taber grinding wheels, with defects that are both homogeneous and widespread [39,40]. In contrast, sample P (Figure 7b) shows a less severe impact: the surface appears smoother and lacks the characteristic abrasion lines from the Taber grinding wheels. Several ceramic microspheres can still be seen, firmly attached to the layer matrix, while multiple holes are visible where LDPE granules were expelled during the abrasion process. Thus, the surface appearance aligns with the Taber test results: sample P shows less damage from abrasive phenomena compared to the reference sample R. This effect may be attributed to a self-lubrication process exerted by the LDPE granules, similar to observations with other polymer-based fillers [41,42]. These granules absorb the shearing forces exerted by the abrasive wheels, thereby reducing their impact on the polymeric matrix of the paint. After all, *PolyTuf^®^ 1229* was specifically engineered to enhance the abrasion resistance of organic coatings. Even seashell powder appears to significantly reduce the impact of the Taber grinding wheels, resulting in less mass loss in sample S. This effect is illustrated by a specific process involving the larger seashell granules in Figure 7c. These wide particles, well integrated into the coating, resist the movement of the abrasive wheels. The presence of ‘steps’ formed between various large granules act as resistance points against the grinding wheel’s movement. Consequently, the wheels tend to glide over these steps, leading to less removal of the polymeric matrix of the coating. Finally, while mussel-derived powders provide positive mechanical reinforcement, their effectiveness is not as pronounced as the other two previous bio-based fillers. The layered structure of mussel powders, as shown in Figure 1c, is less effective in resisting abrasion processes. The Taber grinding wheels exert strong shearing forces that can cause the layers of mussel granules to separate. Once this de-cohesion begins, the powders lose much of their protective capability, despite their demonstrated hardness (Figure 5). Figure 7d shows large amounts of powders on the coating surface, resulting from the fragmentation of previously larger granules. Consequently, part of the mussel powders could act as a third body in the abrasive process, exacerbating the wear phenomena. Thus, the overall performance of the mussel powders falls between that of the pure polymer matrix in sample R and the more effective fillers in samples P and S.

Ultimately, these tests demonstrated that the various fillers, in the quantities investigated in this study, do not significantly affect the surface hardness properties of the coatings. However, they greatly enhance the abrasion resistance of the polymer matrix in the composite layers. Specifically, mussel-derived powder, with its layered structure, is less effective at counteracting the shear stresses of abrasive processes. Conversely, seashell powders, which are very compact, exhibit excellent resistance to the movement of abrasive wheels, making them an outstanding natural alternative to industrial fillers like *PolyTuf^®^ 1229*. Moreover, this material exhibits better behavior to other types of bio-based fillers, such as cellulose nanocrystals [43,44] and olive powders [38], which have highlighted critical issues associated with agglomeration phenomena that reduce the abrasion resistance of paints.

### 3.3. Impact of the Additive on the Coating’s Durability in Aggressive Environments

#### 3.3.1. Salt Spray Chamber Exposure

Polyurethane acrylate coatings exhibit water and chemical resistance, as they have been tested and recommended for use in acidic, neutral, and alkaline environments. If cathodic delamination occurs, the coating itself is not expected to suffer damage; however, the steel/coating interface may be compromised, manifesting as blistering or a “detached crown” around any scratches.

Samples were exposed into a salt fog chamber for 168 h. The reference sample R, lacking inhibitors in the clear coating, displayed rust in the scratch area as illustrated in Figure 8. After 72 h, small blisters began to form around the scratch and developed over time, with the primary damage being filiform corrosion due to chloride presence in the scratch [45,46].

Sample P reinforced with *PolyTuf^®^ 1229* powder showed stable small blistering after 100 h, with less corrosion in the scratch compared to the reference samples. However, delamination around the scratch was similar to that observed with micronized mussel powder (sample M). Coating adhesion is crucial as poor adhesion creates paths for water, oxygen, or ions, accelerating corrosion. Strong adhesion helps prevent delamination due to moisture presence. While micronized shells (sample S) were expected to act as a barrier, they did not provide the anticipated inhibition effect. Rust formation in the scratch and filiform corrosion indicated the creation of preferential paths at the shell powder/resin interface in high humidity. Filiform corrosion filaments often progressed as blisters in these conditions.

Figure 9 illustrates how particle presence reduces metal isolation by forming paths, leading to increased delamination due to cathodic reactions. This increased delamination in shell-aggregate samples may result from the shell structure facilitating oxygen passage, thereby supporting the cathodic reaction.

Therefore, it can be concluded that the inclusion of bio-based fillers, while beneficial for abrasion resistance, is detrimental to the corrosion resistance of the coating. The barrier properties of the composite layers are compromised by the presence of these powders, as their interface with the polymer matrix creates pathways for moisture and aggressive ions to penetrate. After all, similar behavior has been observed in previous studies [38,47], where lignocellulose-based fillers contributed to water uptake due to the formation of percolating pathways that facilitate water transport through capillarity within the filler. Thus, to quantitatively assess the impact of the three bio-based additives on the protective performance of the coatings, the samples were subjected to particularly aggressive UV-B radiation and electrochemical impedance spectroscopy measurements.

#### 3.3.2. UV-B Chamber Exposure

A common issue with using natural and bio-derived fillers into organic coatings is their tendency to undergo physical and chemical deterioration when exposed to sunlight [48,49]. To assess this, the samples were exposed in a UV-B chamber to evaluate any potential aesthetic changes in the coatings.

Figure 10 presents the results of gloss and color measurements analyzed during the accelerated degradation test. Figure 10a illustrates the gloss trends. The reference sample R demonstrates a nearly constant gloss trend with insignificant variations, suggesting that polyurethane-acrylate paint is not particularly impacted by physical-chemical degradation from UV-B radiation. Similarly, the three composite coatings do not show variations in gloss, demonstrating the good physical-chemical resistance of the bio-based fillers to UV-B radiation exposure. Furthermore, this consistency is confirmed by the color variation depicted in Figure 10b: the composite samples exhibit a color change (ΔE) of approximately 1 unit, which is almost negligible. These performances align with the excellent results observed with other bio-based fillers of a polymeric nature, such as polyamide 11 powders [41], and are significantly better than those of lignocellulose fillers, which tend to photo oxidize easily when exposed to UV radiation [38]. In contrast, Sample R shows a greater color change, highlighting the superior stability of the bio-based fillers compared to the pure polyurethane-acrylate matrix. Specifically, observing the variations in the individual color coordinates L*, a*, and b* shown in Figure 10c, Figure 10d, and Figure 10e, respectively, it is evident that these minimal color changes are due to a slight darkening of the coatings (reduction in L*) and slight yellowing (increase in b*).

The UV-B chamber exposure test confirms the effective aesthetic durability of bio-based fillers derived from seashells, owing to the calcite’s resistance to impacting radiations [50]. Consequently, these types of fillers are suitable for outdoor applications exposed to direct solar radiation. However, this finding contrasts with the results from the salt spray chamber exposure tests, where contact with aqueous solutions may compromise the coating’s integrity and protective performance. Thus, to further investigate, the samples were characterized by EIS measurements to evaluate the impact of the additives on the barrier properties of the polyurethane-acrylate matrix.

#### 3.3.3. Electrochemical Impedance Spectroscopy

Electrochemical impedance spectroscopy is a technique that allows the evaluation of highly resistive systems, such as organic, inorganic, or hybrid coatings [51]. The main advantage of using EIS for assessing the performance of organic coatings is that it involves applying only a small polarization to the surfaces, making EIS a non-destructive technique. Polyurethane-acrylate coatings are recommended for industrial environments [52] due to their high stability in water and chemical environments, suggesting high resistance in saline solutions. Amirudin et al. [53] established that a coating is protective if it has a total resistance of at least 10^6^ Ohm*cm^2^, whereas values lower than this indicate a degraded or non-protective coating.

In this research, the total resistance of the coated samples is reported as the modulus of impedance at low frequency, 0.01 Hz, and compared to assess the effect of bio-aggregates in the polymeric matrix (Figure 11). As expected, the reference sample R coated without aggregates maintained high impedance values, between 10^9^ and 10^10^ Ohm*cm^2^, during continuous exposure to saline solution for 168 h. These values are indicative of excellent protective performance. Sample P containing *PolyTuf^®^ 1229* powder also showed high performance throughout the experimental period. The distribution of the aggregates throughout the coating ensured good sealing and isolation of the substrate, thereby improving barrier properties. In fact, the highest total resistance values were observed with *PolyTuf^®^ 1229* powder. Conversely, samples with micronized seashells (sample S and sample M) exhibited low impedance values from the outset, indicating poor performance over time compared to the reference sample. This behavior is likely due to the formation of preferential paths at the particle/coating interface, allowing water to pass through these paths rather than the polymeric matrix [54]. The consistently high impedance values in the reference samples suggest that the poor performance of the seashell samples is not due to water uptake [55]. The structure of the shells, which have a plated-like pattern [9], facilitates the exchange of liquids and gases, potentially influencing long-term exposure depending on particle size.

Ultimately, both the salt spray chamber exposure tests and the EIS measurements revealed critical issues when the composite coating encounters high humidity or liquid solutions. The interface between the bio-based filler and the coating matrix is prone to defects, which facilitate the percolation of ions and aggressive molecules to the metal substrate. Thus, the application of fillers derived from marine shell waste requires optimization for use in outdoor coatings. For example, a bi-layer system could be designed by applying the bio-based fillers only in the topcoat to leverage their superior anti-abrasive properties without compromising the metal substrate.

However, UV-B chamber exposure and subsequent color and gloss measurements indicated that these fillers do not undergo specific physical-chemical degradation. Therefore, with improvements in interface energy between the filler and coating matrix, these bio-based fillers can serve as a sustainable and effective alternative to current synthetic fillers.

## 4. Conclusions

This research investigated the potential use of seashell-derived powders as functional and environmentally sustainable fillers in paint formulations. The study focused on examining their influence on the aesthetic, mechanical, and protective properties of a polyurethane-acrylate paint, comparing the results with those obtained using a commercial composite additive.

The bio-based fillers significantly reduced the gloss levels of the polymer matrix by more than 70 points, introducing a pronounced mattifying effect, yet they did not alter the surface roughness of the coatings. Additionally, these additives caused only a minimal color change, around 4 units. Therefore, while these powders are effective as mattifying agents, they are not suitable as specific pigments.

Additionally, the fillers do not significantly affect the hardness of the coatings but enhance the abrasion resistance characteristics of the composite layers. Notably, seashell powders, due to their compactness, exhibit excellent resistance to abrasive wheel movement, reducing the mass loss in the Taber test by 67% compared to the pure polymer matrix. This makes them an excellent natural alternative to industrial fillers for high abrasion resistance applications.

However, both the salt spray chamber exposure tests and the EIS measurements identified significant issues when the composite coating is exposed to high humidity or liquid solutions. The interface between the bio-based filler and the coating matrix tends to develop defects, allowing ions and aggressive molecules to permeate to the metal substrate. Conversely, exposure to UV-B radiation demonstrated that coatings with bio-based fillers possess excellent physical-chemical durability. Thus, these accelerated degradation tests indicate that the use of seashell-derived fillers for outdoor applications is feasible only after enhancing the interface energy between the filler and coating matrix to minimize the potential for aggressive solution percolation.

In conclusion, the tests conducted in this study have shown that fillers derived from seashell waste, such as shells and mussels, can serve as a viable alternative to modern industrial fillers, thereby reducing the environmental impact of the paint industry. While some issues remain, particularly concerning the protective properties of the composite coating, future research could aim at enhancing these aspects. This would enable the large-scale application of these abundant marine waste-derived fillers in the functional paint industry. Specifically, significant efforts should be directed towards analyzing the effect of filler size, as it could greatly influence the performance of the powder derived from seashells. With the excellent chemical-physical properties of the seashells confirmed, the size of the powders becomes a crucial parameter that can either reduce the protective properties of the coating or enhance its performance, potentially through better homogeneous distribution within the polymeric matrix. Therefore, a thorough analysis of the role of powder size is essential for the correct industrial application of this type of bio-based filler.

## Figures and Tables

**Figure 1 materials-17-04134-f001:**
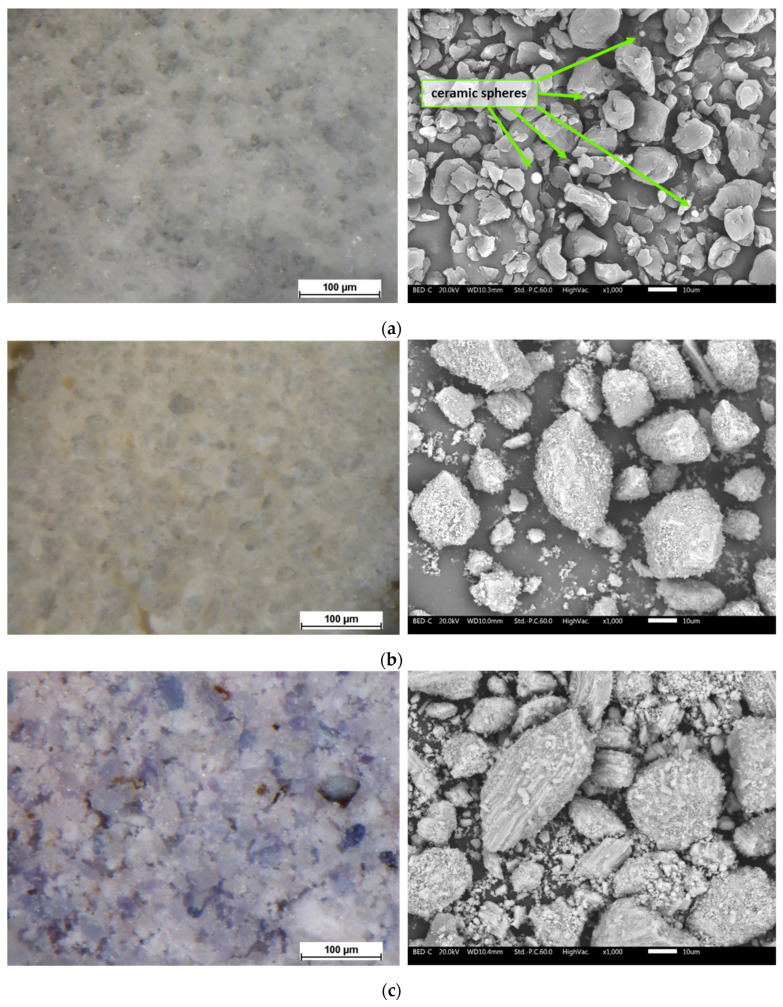
Optical micrographs (**left**) and SEM images (**right**) depicting the appearance and surface morphology of the powders of (**a**) *PolyTuf^®^ 1229*, (**b**) seashells, and (**c**) mussels.

**Figure 2 materials-17-04134-f002:**
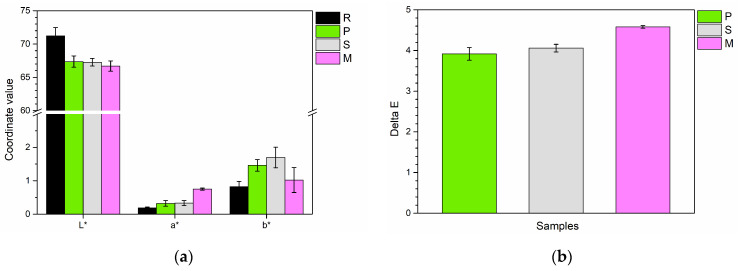
(**a**) The color coordinates L*, a*, and b* for the four samples, and (**b**) the ΔE values of the three composite coatings relative to the reference sample R.

**Figure 3 materials-17-04134-f003:**
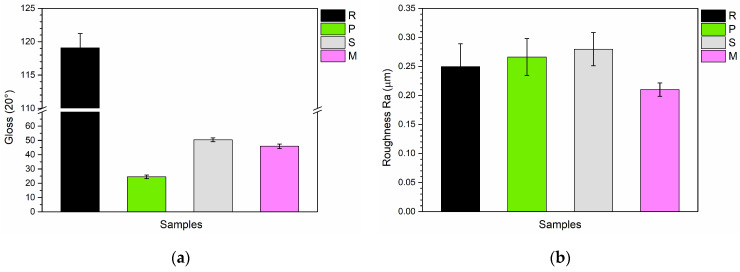
(**a**) Gloss values of the samples and (**b**) the corresponding roughness values.

**Figure 4 materials-17-04134-f004:**
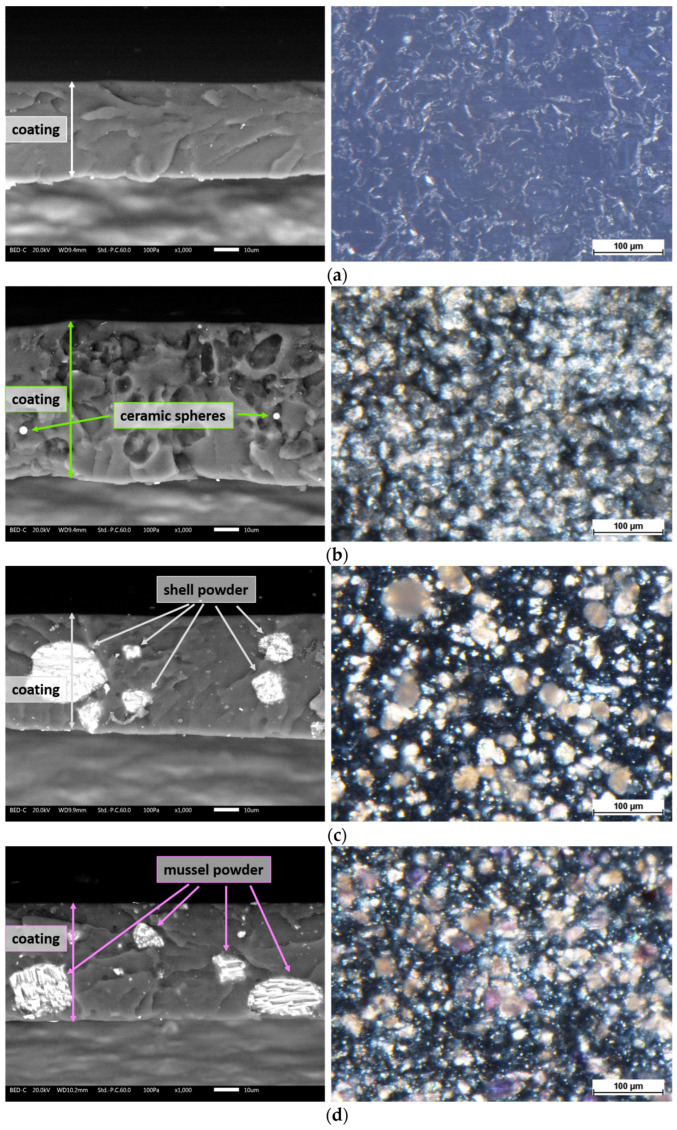
SEM micrographs of the cross-section (on the **left**) and stereomicroscope images of the top-view (on the **right**) of (**a**) sample R, (**b**) sample P, (**c**) sample S and (**d**) sample M.

**Figure 5 materials-17-04134-f005:**
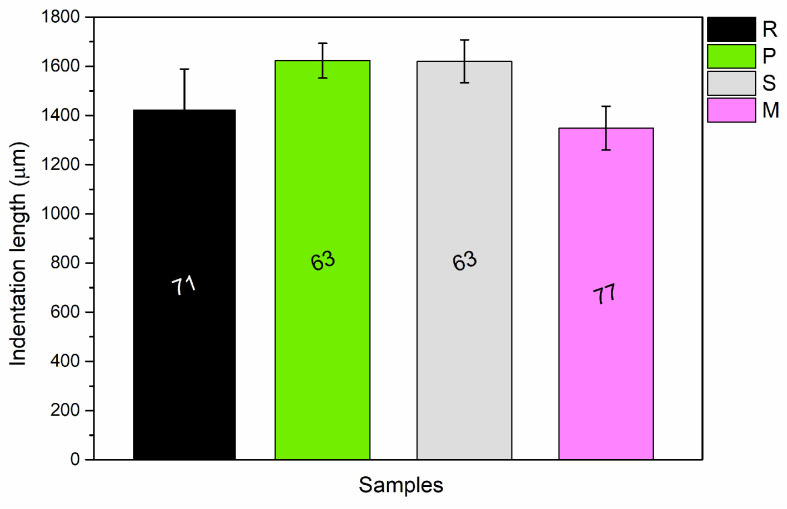
The average dimensions of indentation imprints recorded during Buchholz hardness assessments, along with their corresponding Buchholz hardness values.

**Figure 6 materials-17-04134-f006:**
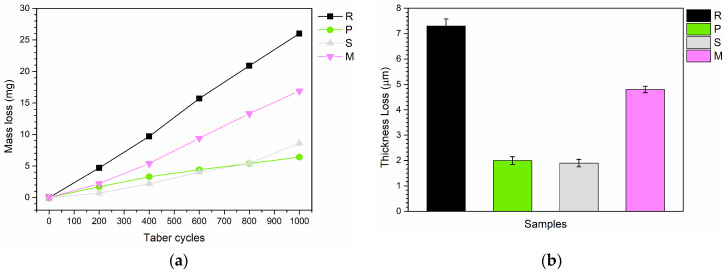
(**a**) Mass loss measured during the Taber test, and (**b**) the reduction in coating thickness after 1000 abrasion cycles.

**Figure 7 materials-17-04134-f007:**
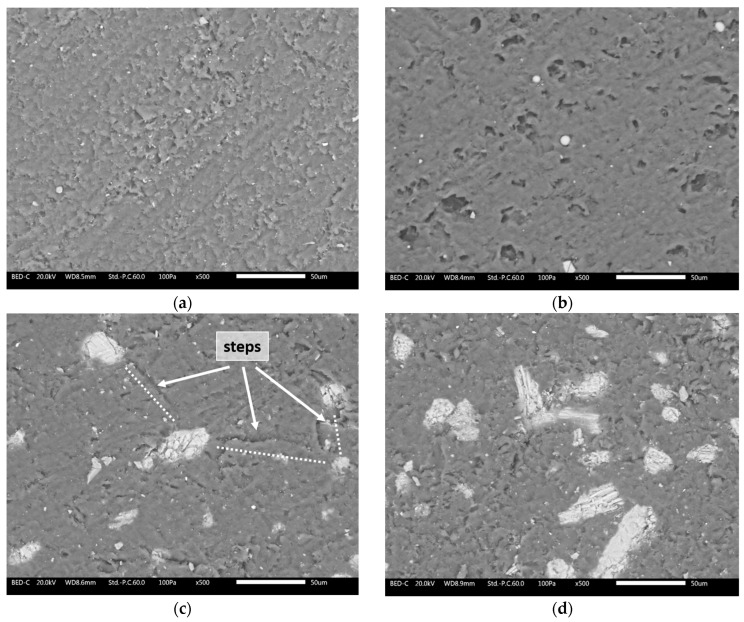
SEM micrographs of the surface morphology of (**a**) sample R, (**b**) sample P, (**c**) sample S, and (**d**) sample M after the Taber test.

**Figure 8 materials-17-04134-f008:**
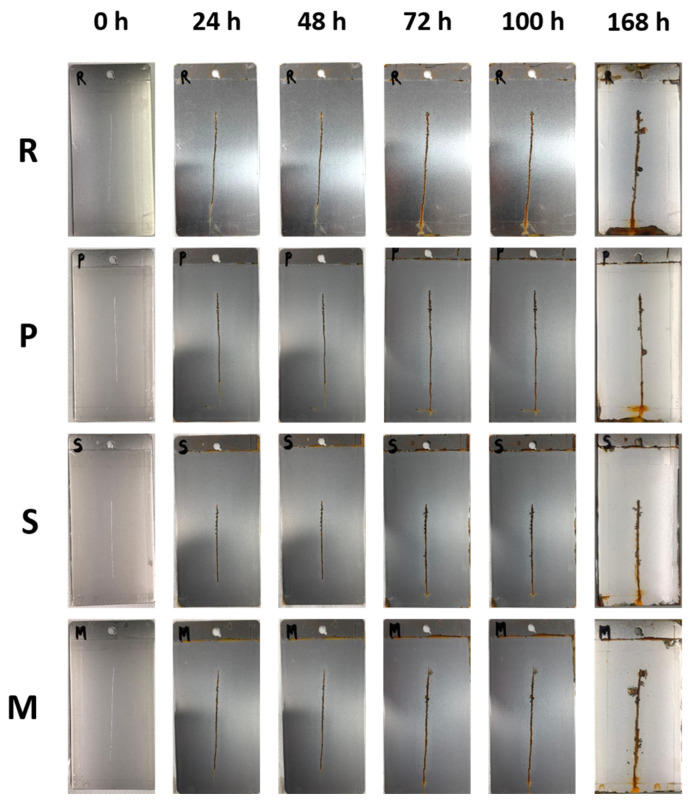
Changes in the appearance of the samples during exposure in a salt spray chamber.

**Figure 9 materials-17-04134-f009:**
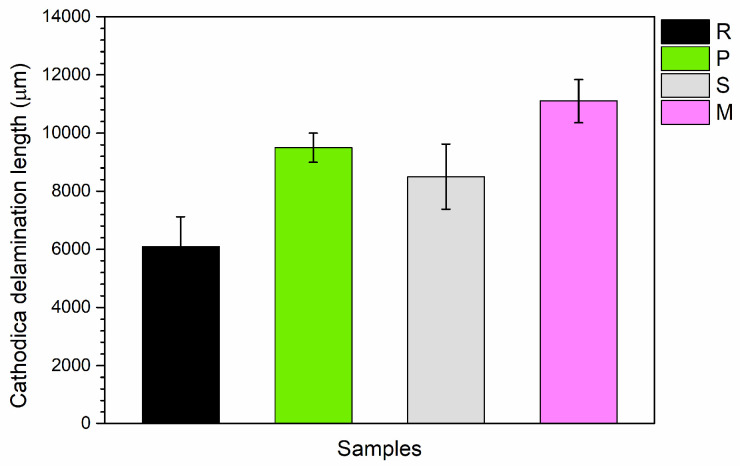
Width of the cathodic delamination front measured at the notch following the exposure test in the salt spray chamber.

**Figure 10 materials-17-04134-f010:**
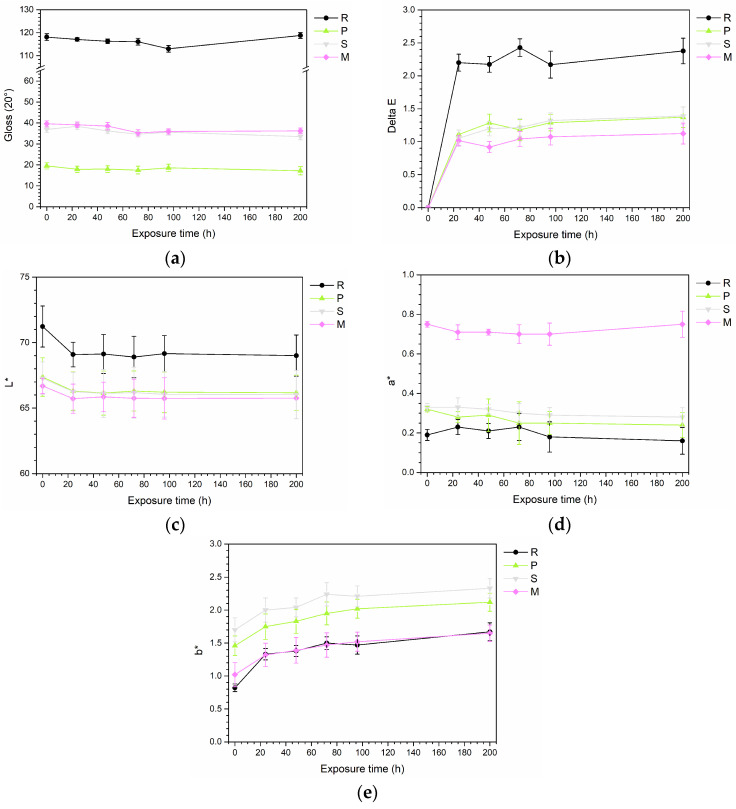
Changes in (**a**) gloss, (**b**) color change, (**c**) L* coordinate, (**d**) a* coordinate, and (**e**) b* coordinate during the UV-B radiation exposure test.

**Figure 11 materials-17-04134-f011:**
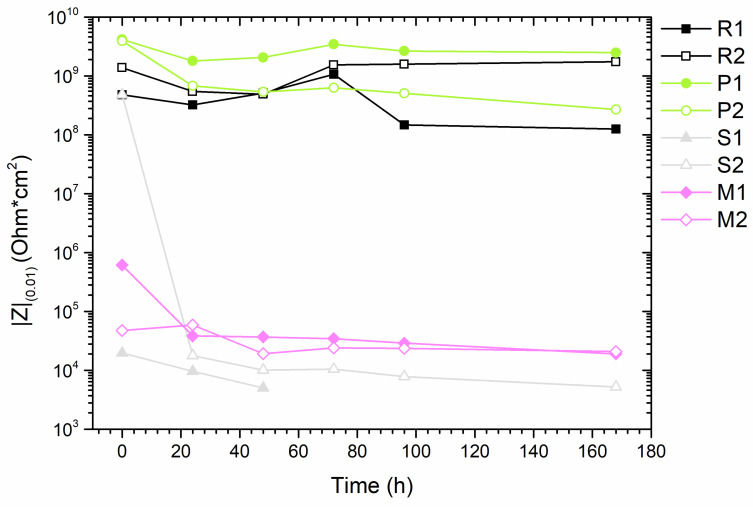
Bode impedance modulus |Z|_(0.01)_ evolution with time.

**Table 1 materials-17-04134-t001:** Sample composition with relative nomenclature.

Samples Nomenclature	Additives (5 wt.%) Introduced in the Paint Formulation
R	/
P	*PolyTuf^®^ 1229*
S	Seashell powder
M	Mussels powder

## Data Availability

The data presented in this study are available on request from the corresponding author. The data are not publicly available due to the absence of an institutional repository.

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
