# Peer review of "Micronized Shell-Bioaggregates as Mechanical Reinforcement in Organic Coatings"

_materials, 2024, doi:10.3390/ma17164134_

Round 1

Reviewer 1 Report

Comments and Suggestions for Authors

The topic is interesting regarding the possible use of the micronized shell bioaggregates first of all because of the possible large-scale application.

There are some remarks:

1. line 85: please describe more detailed the grinding process! It is important for the resulting powder size and morphology!

2. line 107: please describe more precise the used spray method!

Conclusions: the authors should explain the key parameters, which mostly influence the quality of the coatings in the case of the seashell waste!

2. 

Author Response

The topic is interesting regarding the possible use of the micronized shell bioaggregates first of all because of the possible large-scale application.

There are some remarks:

  1. line 85: please describe more detailed the grinding process! It is important for the resulting powder size and morphology!

Authors: the authors thank the reviewer for the useful advice, and have added the following sentences on line 86: “Specifically, an MGS ball mill (MGS, Fiorano Modenese, Italy) was used to grind the powders. The process utilized aluminous porcelain spheres with a diameter of 1 cm, which were stirred inside the jar containing the powder for a duration of 10 minutes.”

  1. line 107: please describe more precise the used spray method!

Authors: the authors have already introduced online 108 all the information related to the spraying process, such as pressure and rates of applied material: “Thus, the various paint formulations were applied using a spray method with pressure of 3 bar, achieving a coverage rate of approximately 8 m²/L.”

Conclusions: the authors should explain the key parameters, which mostly influence the quality of the coatings in the case of the seashell waste!

Authors: the authors believe that the size of the powders represents a determining aspect on their performance within the coating matrix. Consequently, the authors have added the following sentences at the end of the Conclusions: “Specifically, significant efforts should be directed towards analyzing the effect of filler size, as it could greatly influence the performance of the powder derived from seashells. With the excellent chemical-physical properties of the seashells confirmed, the size of the powders becomes a crucial parameter that can either reduce the protective properties of the coating or enhance its performance, potentially through better homogeneous distribution within the polymeric matrix. Therefore, a thorough analysis of the role of powder size is essential for the correct industrial application of this type of bio-based filler.”

Reviewer 2 Report

Comments and Suggestions for Authors

The paper talks about the incorporation of biobased fillers into  coating system. Authors have incorporated these fillers and studied their mechanical performance.

1. Authors used 60 degree C for 40 minutes as curing condition for all samples. Did authors do DSC to make sure that the curing condition is same for all the fillers? Fillers can affect the curing condition and curing condition is crucial while comparing performance.

2. Quality of figure 10 can be improved

3. Authors talk about incorporation biobased fillers in the coating system. Whenever such research is presented it is advisable to add reviews that discuss the biobased standards and ASTM test to ensure the credibility of such claims. eg. https://doi.org/10.3390/su152215758

4. Is there a reason for selecting this particular  polyurethane-acrylate paint? Have authors done something similar using different type of paint and if so did they get similar results?

5. It is also advised that authors compare and contrast analogues research articles. eg. https://doi.org/10.1016/j.indcrop.2023.116326 

Author Response

The paper talks about the incorporation of biobased fillers into  coating system. Authors have incorporated these fillers and studied their mechanical performance.

  1. Authors used 60 degree C for 40 minutes as curing condition for all samples. Did authors do DSC to make sure that the curing condition is same for all the fillers? Fillers can affect the curing condition and curing condition is crucial while comparing performance.

Authors: the authors understand the importance of the reviewer's comment. However, they did not perform DSC measurements to confirm the correct cross-linking of the coatings, as the 5 wt.% additive introduced is not significant enough to notably influence the cross-linking process. Additionally, the authors exercised caution by adhering to the paint manufacturer's recommended range of times and temperatures (30-40 minutes at 50-60 °C). Based on their experience with previous works, they opted to use the higher end of these time and temperatures to ensure thorough cross-linking of the samples.

  1. Quality of figure 10 can be improved

Authors: To be sure, the graphs in Figure 10 are in very high quality (600 dpi). They probably appear of lower quality once inserted into the word template. Just in case, the authors will also upload the figures in high quality into the system, so that they appear in the best possible way in the final product.

  1. Authors talk about incorporation biobased fillers in the coating system. Whenever such research is presented it is advisable to add reviews that discuss the biobased standards and ASTM test to ensure the credibility of such claims. eg. https://doi.org/10.3390/su152215758

Authors: the authors disagree with the reviewer's comment. In fact, the authors have already spoken extensively in the introduction about the importance of using bio-based additives in proactive coatings, which represents a real current challenge: “Nevertheless, the use of bio-based materials often poses significant challenges related to the protective performance, durability, and color consistency of additives in the paint [8]. The sea represents a significant resource of bio-based fillers that can be effectively utilized. Molluscs' seashell waste, abundant in calcium carbonate (CaCO3) as its major component, offers a renewable alternative to conventional sources like calcium carbonate. Specifically, marine shells such as Acanthocardia tuberculata and Mytilus galloprovincialis are predominantly composed of calcite and aragonite [9]. Calcite derived from these shells can serve as a filler in organic coatings and has been studied for its potential to reduce the need for TiO2 [10]. Utilizing calcite from marine shells not only reduces coating costs but also enhances mechanical properties. Research has explored using marble residues [11] to promote circular economy practices and employ environmentally sustainable materials. Recent studies have shown that calcium carbonate obtained from seashells can enhance the mechanical [12,13] and thermal [14] properties of bio-composites. Additionally, it improves the durability and enhances flame retardancy and smoke suppression in coatings [15,16]. These findings underscore the multifaceted benefits of utilizing seashell-derived calcium carbonate in various applications, highlighting its potential as a valuable bio-based material.”

Furthermore, the bio-based nature of the two types of powders derived from marine shells is truly evident, as they are manually obtained from sea products. Their green credentials are unquestionable, as the powders have only been subjected to dry grinding of the original shells without any additional treatment.

Finally, the recommended review is considered off-topic, as it addresses considerations on microplastics and the sustainability of polymers. In contrast, this work focuses on bio-based materials derived from marine waste, which are unrelated to synthetic or natural polymeric materials.

  1. Is there a reason for selecting this particular  polyurethane-acrylate paint? Have authors done something similar using different type of paint and if so did they get similar results?

Authors: The authors thank the reviewer for the important comment. They have not studied the effect of this type of powder in other paints but chose this specific polyurethane-acrylate paint for several reasons. First, as a clear coat, it is free of additives and pigments that could reduce or influence the effect of the bio-based fillers characterized in the study. Furthermore, being transparent, the paint could facilitate the observation of the fillers introduced inside it. Lastly, previous studies have demonstrated the good outdoor durability of this paint, which helps in better evaluating the durability of fillers in accelerated degradation tests.

Therefore, the authors added the following sentences in line 95: “This specific polyurethane-acrylate paint was used for several reasons. First, as a clear coat, it is free of additives and pigments that could reduce or influence the effect of the bio-based fillers characterized in the study. Additionally, the transparency of the paint facilitates the observation of the fillers introduced into it. Finally, its good outdoor durability allows for the evaluation of the fillers' behavior in accelerated degradation tests.”

  1. It is also advised that authors compare and contrast analogues research articles. eg. https://doi.org/10.1016/j.indcrop.2023.116326 

Authors: the authors have added several analyzes and comparisons with literature works in the text. For example, on line 293: “On the other hand, the literature demonstrates the difficulty of improving the hard-ness of paints using bio-based fillers [35,36], unless they are derived from very hard materials, such as macadamia nuts [37] or olive stones [38].” On line 366: “Moreover, this material exhibits better behavior to other types of bio-based fillers, such as cellulose nanocrystals [43,44] and olive powders [38], which have highlighted critical issues associated with agglomeration phenomena that reduce the abrasion resistance of paints.” On line 407: “After all, similar behavior has been observed in previous studies [38,47], where lignocellulose-based fillers contributed to water uptake due to the formation of percolating pathways that facilitate water transport through capillarity within the filler.” On line 427: “These performances align with the excellent results observed with other bio-based fill-ers of a polymeric nature, such as polyamide 11 powders [41], and are significantly better than those of lignocellulose fillers, which tend to photo oxidize easily when ex-posed to UV radiation [38].”